# Vehicular Emission: Estimate of Air Pollutants to Guide Local Political Choices. A Case Study

**Sergio Mazza [1], Donatella Aiello [2]** **, Anastasia Macario [1]** **and Pierantonio De Luca [3],***

[1] Department of Ingegneria dell'Ambiente, University of Calabria, I-87030 Arcavacata di Rende (CS), Italy; sergiomazza87@gmail.com (S.M.); anastasia.macario@unical.it (A.M.)

[2] Department of Chimica e Tecnologie chimiche, University of Calabria, I-87030 Arcavacata di Rende (CS), Italy; donatella.aiello@unical.it

[3] Department of Ingegneria Meccanica, Energetica e Gestionale, University of Calabria, I-87030 Arcavacata di Rende (CS), Italy

\* Correspondence: pierantonio.deluca@unical.it; Tel.: +39-0984-496757

**Abstract:** The aim of this case study was to show how, with the use of software, is it possible to carry out a preventive screening of vehicular emissions. Moreover, thanks to this preliminary analysis, some areas that are potentially polluted can be identified in advance and suitable samplings on small-scale on them would help to verify the effectiveness of policies that can be adopted for the reduction of pollution. To this end, this paper reports a case study on vehicle traffic pollution in Calabria, a region in the south of Italy. We used the methodology called Corinair (Coordination Information AIR), developed by the EEA (European Environment Agency) and uses the software Copert4 (Computer Program to calculate Emission from Road Traffic). The total emissions per area were analyzed and the emissions for particular pollutants per unit area ($km^2$) and per citizen were considered. The obsolete vehicles determined a substantial impact on the local atmospheric pollution. It was demonstrated how it is possible to substantially reduce the pollution of an area by adopting policies that encourage, for example, through tax concessions, the replacement of old cars of private citizens.

**Keywords:** Corinair; particulate matter; pollution; traffic pollution

## 1. Introduction

The state of air pollution is the result of a process that includes all the phenomena in which pollutants take part, starting from their genesis and up to the completion of their negative action on the ecosystem. The problem of air pollution is mainly concentrated in metropolitan areas, where vehicular traffic, industrial plants, and building heating have detrimental effects on air quality and the health of inhabitants [1]. In addition, natural aerosols can significantly affect air quality, even in anthropized areas [2].

The most common pollutants in the atmosphere are sulfur dioxide ($SO_2$), nitrogen oxides (NOx), carbon monoxide (CO), benzene, polycyclic aromatic hydrocarbons (PAH) [3–5], and atmospheric particulate matter (PM), which is one of the most dangerous pollutants for humans and is most common in cities [6–9]. The latter comes mainly from vehicular traffic, which produces about a quarter of total emissions in urban air [10–12].

Cars, through the combustion process, generate the pollutants that are released into the atmosphere. A key variable in the process is the type of combustion and the fuel used. Engines created with obsolete technologies will have a far greater contribution than innovative engines [13–17].

Today, the increased demand for cars for private use has led to a 2.8% annual increase in production in the automotive sector [18]. The emissions due to the ever-increasing number of cars are, however, fortunately reduced by the scientific research of new materials, which can also be used in catalytic converters and are capable of absorbing [19–27] and photodegrading [28–30] pollutant products. Of particular interest in recent years is the research directed at identifying bioindicators to assess the effect of pollutants on the environment [31,32].

In Europe, a significant part of the European population live in areas, especially in cities, where the limits set by air quality standards are exceeded. Although the emissions of many air pollutants have decreased substantially in recent decades, thus leading to an improvement in air quality, the concentrations of many pollutants continue to be too high [33–35].

In recent decades, several laws have been passed to force manufacturers to reduce the lead content first to the point of canceling it as well as benzene as both are highly carcinogenic [36].

The reduction of air pollution has been addressed on several levels through research, but also through legislation, cooperation with the sectors responsible for air pollution, and with international, national, and regional authorities. The adoption of political actions to combat air pollution, which must be shared internationally, can only be satisfied through a synergy of multiple actions that must be carried out at the national level and on a small scale.

The long-term goal of the EU is to achieve air quality levels that have no consequences for human health and the environment. By 2030, according to the Paris Treaty of 2015, each country must reduce its polluting emissions by 40% [37]. Road traffic represents a determining source of air pollution in urban areas [38]. For this reason, it is important to quantify road traffic emissions and try to reduce them. Most European countries have accepted and adopted [39–41] a methodology capable of creating a database for estimating the emissions of air pollutants due to road transport. Italian regions and provinces are required to implement regional and provincial inventories for the preliminary assessment of air quality and the implementation of urban traffic plans. This work reports a case study conducted in Calabria, a region of southern Italy, where the Corinair methodology was applied to estimate the emission of pollutants produced by vehicle traffic only and where, to our knowledge, there are no data relating to the monitoring of vehicle traffic emissions covering the entire study area. This study area was chosen because it is characterized by low industrialization and therefore vehicle traffic emissions particularly contribute to air quality and consequently, the estimate of the latter can be especially useful.

Finally, an estimate was made of the reduction for some particular pollutants assuming particular scenarios.

## 2. Materials and Methods

The case study was carried out to quantify and study the emissions of pollutants resulting from vehicular traffic in the Calabria region using the Corinair methodology [39–41]. This methodology is able to create a database of emissions thanks to the software Copert 4 (COmputer Program to calculate the Emission from Road Traffic).

It is a tool that is able to quantify the emissions of each pollutant produced by vehicles and has a good reliability thanks to its huge database that includes every type of motor that circulates in Europe and every type of fuel. The software also requires some fundamental data, related to the year and the municipality considered such as climatic data, average fuel consumption and its characteristics, features of the car fleet, and data on vehicle mileage and road features (e.g., average slope) [39].

In particular, in this study case, climate data were found in the online database of ArpaCal (regional agency for environmental protection of Calabria).

Data on fuels were obtained from the online database of the Ministry of Economic Development [42] The data relating to the circulating fleet were available on the ACI database (Automobile Club Italia), considering that the entire calendar year of 2015 was available as a complete set of data. With regard to Calabria, it was possible to obtain all the data relating to the fleet, divided by province and municipality. The database, in the public domain, divides the vehicles according to the type: cars; light commercial

vehicles (<3.5 t); heavy commercial vehicles (>3.5 t); buses, motorcycles, etc. It further divides the vehicles by engine size and fuel (petrol, diesel, methane, LPG (liquid petroleum gas), electricity). A final classification concerns vehicle technology: Euro 0; Euro 1; Euro 2; Euro 3; Euro 4; Euro 5; and Euro 6. Technology is essential as it provides a lot of information such as year of registration, fuel consumption, engine wear, conditions, and pollutant emissions. There is no doubt that a Euro 0 vehicle will have obsolete technology compared to a Euro 4, and its emissions will be greater than a new vehicle. Unfortunately, knowing the average slopes on a vast territory like an entire province is almost impossible. In order to cope with this, Copert suggests percentages that are averages of the European territory. Once all the parameters were known, it is possible to proceed with the calculation of the emission factors and all the parameters necessary for the evaluation of the emissions of pollutants. The work was carried out in different and consecutive phases. First, the vehicular emissions of CO (carbon monoxide), PM (particulate matter), VOC (volatile organic compounds), and $NO_2$ (nitrogen oxide) were investigated across the whole region considering the five provinces of Cosenza, Catanzaro, Reggio Calabria, Crotone and Vibo Valentia. Table 1 shows the characteristics of the five provinces.

**Table 1.** Characteristics of the five Calabrian provinces referring to the year 2015.

| Provinces | Area (Km$^2$) | Number of Inhabitants | Number of Vehicles | Population Density (Inhabitants/Km$^2$) | Vehicles Density (Vehicles/km$^2$) |
|---|---|---|---|---|---|
| Catanzaro | 2391 | 363,979 | 300,471 | 152.23 | 125.67 |
| Cosenza | 6710 | 731,649 | 557,631 | 109.04 | 83.10 |
| Crotone | 1717 | 171,863 | 122,526 | 100.10 | 71.36 |
| Reggio Calabria | 3183 | 559,675 | 431,103 | 175.83 | 135.44 |
| Vibo Valentia | 1139 | 162,697 | 127,909 | 142.84 | 112.30 |

The car parks in the provinces are composed of an important number of obsolete motor vehicles. The province with the largest fleet is that of Cosenza, while the smallest are the provinces of Vibo Valentia and Crotone. Figure 1 shows the number of vehicles in car parks in the five Calabrian provinces distinguished by engine technology.

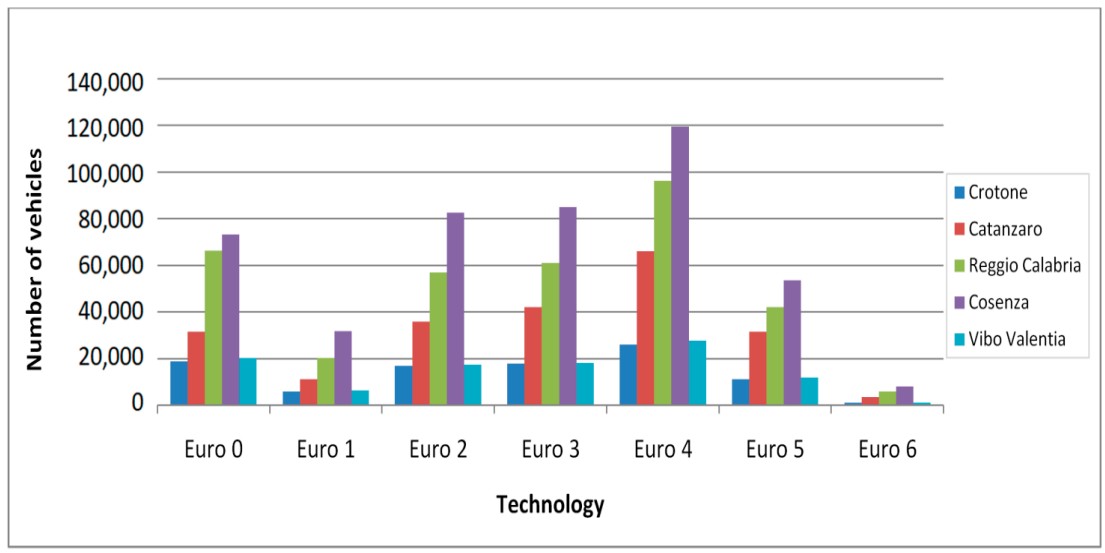

**Figure 1.** Number of vehicles in the car parks of the five Calabrian provinces distinguished by engine technology.

Subsequently, the study focused on analyzing a single province to avoid processing a lot of data. The province of Catanzaro was chosen as it is the capital of the Calabria region and therefore considered

as the most representative province. The study continued on a smaller municipal scale, considering all eighty municipalities in the province of Catanzaro alone (Figure 2 and Table 2).

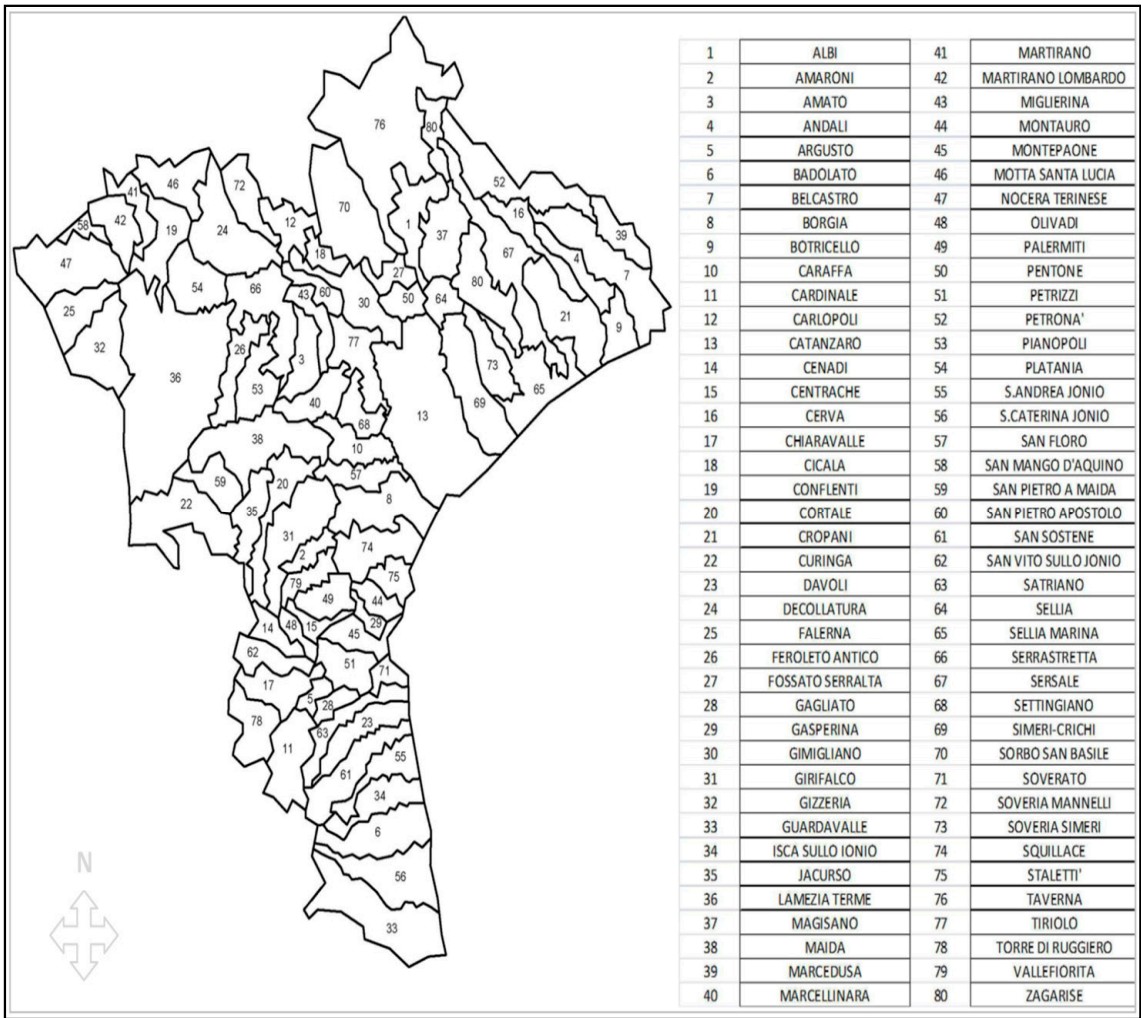

| | | | |
|---|---|---|---|
| 1 | ALBI | 41 | MARTIRANO |
| 2 | AMARONI | 42 | MARTIRANO LOMBARDO |
| 3 | AMATO | 43 | MIGLIERINA |
| 4 | ANDALI | 44 | MONTAURO |
| 5 | ARGUSTO | 45 | MONTEPAONE |
| 6 | BADOLATO | 46 | MOTTA SANTA LUCIA |
| 7 | BELCASTRO | 47 | NOCERA TERINESE |
| 8 | BORGIA | 48 | OLIVADI |
| 9 | BOTRICELLO | 49 | PALERMITI |
| 10 | CARAFFA | 50 | PENTONE |
| 11 | CARDINALE | 51 | PETRIZZI |
| 12 | CARLOPOLI | 52 | PETRONA' |
| 13 | CATANZARO | 53 | PIANOPOLI |
| 14 | CENADI | 54 | PLATANIA |
| 15 | CENTRACHE | 55 | S.ANDREA JONIO |
| 16 | CERVA | 56 | S.CATERINA JONIO |
| 17 | CHIARAVALLE | 57 | SAN FLORO |
| 18 | CICALA | 58 | SAN MANGO D'AQUINO |
| 19 | CONFLENTI | 59 | SAN PIETRO A MAIDA |
| 20 | CORTALE | 60 | SAN PIETRO APOSTOLO |
| 21 | CROPANI | 61 | SAN SOSTENE |
| 22 | CURINGA | 62 | SAN VITO SULLO JONIO |
| 23 | DAVOLI | 63 | SATRIANO |
| 24 | DECOLLATURA | 64 | SELLIA |
| 25 | FALERNA | 65 | SELLIA MARINA |
| 26 | FEROLETO ANTICO | 66 | SERRASTRETTA |
| 27 | FOSSATO SERRALTA | 67 | SERSALE |
| 28 | GAGLIATO | 68 | SETTINGIANO |
| 29 | GASPERINA | 69 | SIMERI-CRICHI |
| 30 | GIMIGLIANO | 70 | SORBO SAN BASILE |
| 31 | GIRIFALCO | 71 | SOVERATO |
| 32 | GIZZERIA | 72 | SOVERIA MANNELLI |
| 33 | GUARDAVALLE | 73 | SOVERIA SIMERI |
| 34 | ISCA SULLO IONIO | 74 | SQUILLACE |
| 35 | JACURSO | 75 | STALETTI' |
| 36 | LAMEZIA TERME | 76 | TAVERNA |
| 37 | MAGISANO | 77 | TIRIOLO |
| 38 | MAIDA | 78 | TORRE DI RUGGIERO |
| 39 | MARCEDUSA | 79 | VALLEFIORITA |
| 40 | MARCELLINARA | 80 | ZAGARISE |

**Figure 2.** Municipalities of the province of Catanzaro.

For the latter it was possible, from the data obtained, to create choropleth maps for each pollutant considered. The total emissions of CO, PM, VOC, and $NO_2$ pollutants per unit area ($km^2$) was considered.

Finally, the percentage reduction in emissions was estimated by supposing a new scenario in which all vehicles in the province of Catanzaro were related to 2015, and equipped with Euro 0 technology were instead with a Euro 1 engine. In this last case, the pollutants studied were CO (carbon monoxide), VOC (volatile organic compounds), NMVOC (non-methane volatile organic compounds), $CH_4$ (methane), $NO_x$ (nitrogen oxides), $NH_3$ (ammonia), PM (particulate matter), $CO_2$ (carbon dioxide), and $SO_2$, as they are the considered determinants in emissions from vehicular traffic.

**Table 2.** Area, number of vehicles, and density of vehicles (number of vehicle/area), in 2015 for cities in the province of Catanzaro.

| n | City | Vehicles | Surface [km²] | Density [Vehicles/Surface] | n | City | Vehicles | Surface [km²] | Density [Vehicles/Surface] |
|---|---|---|---|---|---|---|---|---|---|
| 1 | ALBI | 624 | 29.64 | 21.05 | 41 | MARTIRANO | 609 | 14.57 | 41.80 |
| 2 | AMARONI | 1371 | 9.88 | 138.77 | 42 | MARTIRANO LOMBARDO | 297 | 19.00 | 15.63 |
| 3 | AMATO | 661 | 20.00 | 33.05 | 43 | MIGLIERINA | 524 | 13.00 | 40.31 |
| 4 | ANDALI | 516 | 17.00 | 30.35 | 44 | MONTAURO | 1435 | 11.50 | 124.78 |
| 5 | ARGUSTO | 399 | 7.00 | 57.00 | 45 | MONTEPAONE | 4911 | 16.90 | 1.00 |
| 6 | BADOLATO | 2547 | 34.10 | 74.69 | 46 | MOTTA SANTA LUCIA | 526 | 25.00 | 21.04 |
| 7 | BELCASTRO | 1497 | 52.00 | 28.79 | 47 | NOCERA TERINESE | 3388 | 46.20 | 73.33 |
| 8 | BORGIA | 5225 | 42.00 | 124.40 | 48 | OLIVADI | 381 | 7.00 | 54.43 |
| 9 | BOTRICELLO | 3570 | 15.48 | 230.62 | 49 | PALERMITI | 957 | 18.00 | 53.17 |
| 10 | CARAFFA | 1671 | 25.02 | 66.79 | 50 | PENTONE | 1310 | 12.00 | 109.17 |
| 11 | CARDINALE | 1518 | 30.12 | 50.40 | 51 | PETRIZZI | 782 | 21.00 | 37.24 |
| 12 | CARLOPOLI | 1145 | 16.41 | 69.77 | 52 | PETRONA' | 1972 | 45.50 | 43.34 |
| 13 | CATANZARO | 74,178 | 102.30 | 725.10 | 53 | PIANOPOLI | 3201 | 24.00 | 133.38 |
| 14 | CENADI | 431 | 11.92 | 36.16 | 54 | PLATANIA | 1535 | 24.00 | 63.96 |
| 15 | CENTRACHE | 329 | 7.96 | 41.33 | 55 | S.ANDREA JONIO | 1403 | 20.00 | 70.15 |
| 16 | CERVA | 772 | 21.37 | 36.13 | 56 | S.CATERINA JONIO | 1460 | 41.00 | 35.61 |
| 17 | CHIARAVALLE | 4214 | 23.83 | 176.84 | 57 | SAN FLORO | 456 | 18.00 | 25.33 |
| 18 | CICALA | 662 | 9.22 | 71.80 | 58 | SAN MANGO D'AQUINO | 1117 | 7.00 | 159.57 |
| 19 | CONFLENTI | 1177 | 29.34 | 40.12 | 59 | SAN PIETRO A MAIDA | 3119 | 16.30 | 191.35 |
| 20 | CORTALE | 1560 | 30.01 | 51.98 | 60 | SAN PIETRO APOSTOLO | 1257 | 11.51 | 109.21 |
| 21 | CROPANI | 3358 | 43.00 | 78.09 | 61 | SAN SOSTENE | 1032 | 31.00 | 33.29 |
| 22 | CURINGA | 5043 | 51.47 | 97.98 | 62 | SAN VITO SULLO JONIO | 1243 | 17.00 | 73.12 |
| 23 | DAVOLI | 4401 | 25.70 | 171.25 | 63 | SATRIANO | 2627 | 22.00 | 119.41 |
| 24 | DECOLLATURA | 2468 | 50.35 | 49.02 | 64 | SELLIA | 322 | 12.81 | 25.14 |
| 25 | FALERNA | 3018 | 23.90 | 126.28 | 65 | SELLIA MARINA | 6021 | 40.90 | 147.21 |
| 26 | FEROLETO ANTICO | 1958 | 21.00 | 93.24 | 66 | SERRASTRETTA | 2356 | 41.00 | 57.46 |
| 27 | FOSSATO SERRALTA | 374 | 11.85 | 31.56 | 67 | SERSALE | 3248 | 53.00 | 61.28 |
| 28 | GAGLIATO | 430 | 7.04 | 61.08 | 68 | SETTINGIANO | 2421 | 14.00 | 172.93 |
| 29 | GASPERINA | 1507 | 6.87 | 219.36 | 69 | SIMERI-CRICHI | 1203 | 46.70 | 25.76 |
| 30 | GIMIGLIANO | 2458 | 33.55 | 73.26 | 70 | SORBO SAN BASILE | 591 | 58.00 | 10.19 |
| 31 | GIRIFALCO | 4241 | 43.10 | 98.40 | 71 | SOVERATO | 7129 | 7.70 | 925.84 |
| 32 | GIZZERIA | 3931 | 35.90 | 109.50 | 72 | SOVERIA MANNELLI | 2721 | 20.00 | 136.05 |
| 33 | GUARDAVALLE | 3592 | 60.40 | 59.47 | 73 | SOVERIA SIMERI | 3644 | 22.28 | 163.55 |
| 34 | ISCA SULLO IONIO | 1138 | 22.00 | 51.73 | 74 | SQUILLACE | 2420 | 34.33 | 70.49 |
| 35 | JACURSO | 425 | 21.00 | 20.24 | 75 | STALETTI' | 1760 | 12.11 | 145.33 |
| 36 | LAMEZIA TERME | 53,407 | 162.40 | 328.86 | 76 | TAVERNA | 1664 | 131.22 | 12.68 |
| 37 | MAGISANO | 908 | 31.00 | 29.29 | 77 | TIRIOLO | 2880 | 28.89 | 99.69 |
| 38 | MAIDA | 4288 | 58.20 | 73.68 | 78 | TORRE DI RUGGIERO | 851 | 35.37 | 24.06 |
| 39 | MARCEDUSA | 338 | 15.68 | 21.56 | 79 | VALLEFIORITA | 3326 | 18.83 | 176.63 |
| 40 | MARCELLINARA | 2000 | 20.00 | 100.00 | 80 | ZAGARISE | 1257 | 48.00 | 26.19 |

Density [Number of Vehicles / Surface]

0　　　　　　　　　　　　　　　　1,000

## 3. Results and Discussion

### 3.1. Emissions from Vehicular Traffic, on a Provincial Scale, Extended to the Whole Calabria Region

The following graphs in Figure 3 report the total emissions for 2015, expressed in tons, produced by each province in Calabria.

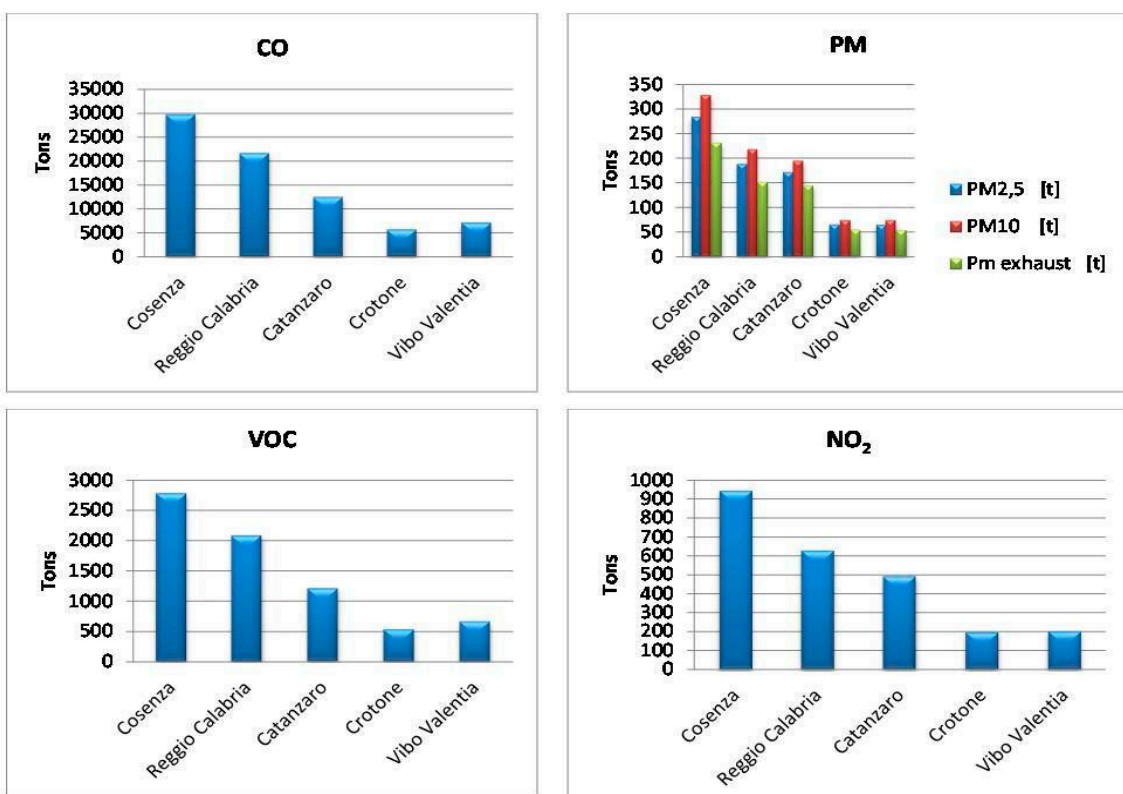

**Figure 3.** Total emissions produced, expressed in tons, in 2015, in the five provinces of Calabria.

As expected, Cosenza, being the province with a greater extension and number of inhabitants, appeared to be the province with the highest emissions of pollutants. The data assume a completely different meaning if they are related to the number of inhabitants, as shown in the graphs in Figure 4.

The results obtained suggest that a careful correlation analysis between the different parameters reverses the situation, as in this case, where the provinces that produced less total amounts of pollution instead had the highest emissions of pollution per inhabitant. This is the case, for example, of Vibo Valentia, which, despite being the smallest province in terms of extension and per inhabitant, had the highest emissions of pollution per inhabitant. This result can be attributable to an "older" fleet in this province relating to the number of inhabitants; consequently, the emissions are higher than those of a province that has a newer and more performing fleet. All this shows how the application of the Corinair methodology allows for the estimation of emissions and the analysis of data according to the context.

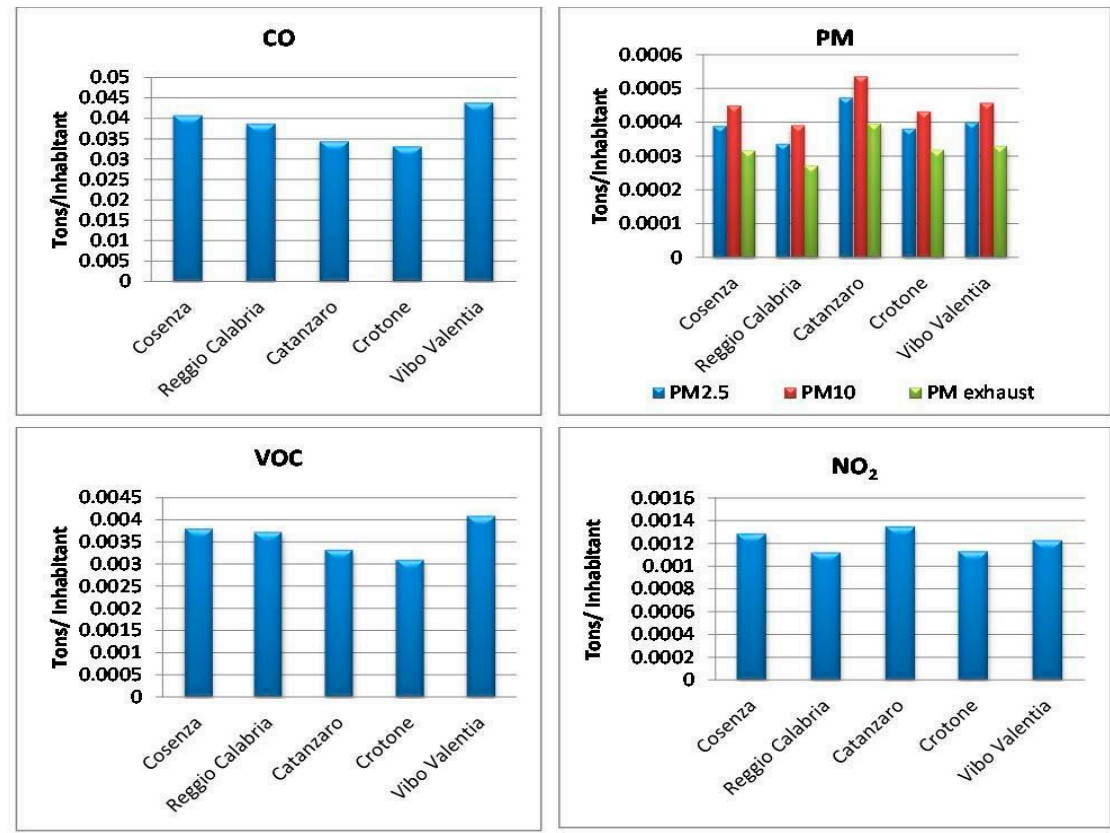

**Figure 4.** Emissions produced, expressed in tons/inhabitant, in 2015, in the five Calabrian provinces.

*3.2. Emissions from Vehicular Traffic, on a Municipal Scale, Extended Only to the Province of Catanzaro*

Figure 5 shows the choropleth maps expressed for the total tons produced in 2015. Vehicle emissions have been associated with particular geographical areas, considering that vehicles are only registered in the corresponding municipality. Although on such a small scale, it may be plausible that there is a circulation of vehicles, it was assumed that the latter circulated within the municipality of registration.

The data reported in Figure 5 show that the municipalities with a higher total production of pollutants were Catanzaro (*n* = 13 in Table 2) and Lamezia Terme (*n* = 36 in Table 2). This is due to the fact that the two centers have bigger car parks. It is, however, necessary to underline that the emissions, even if much higher than those of the small municipalities, are always limited when compared to the average of the large national centers. It should be added that this study returned the annual total of emissions, which is not regulated by law. The legislation imposes limits on the hourly and daily concentrations, but not on the total emissions. A different conclusion is reached if the data reported in Figure 5 are expressed per unit area (Km$^2$), as shown in Figure 6.

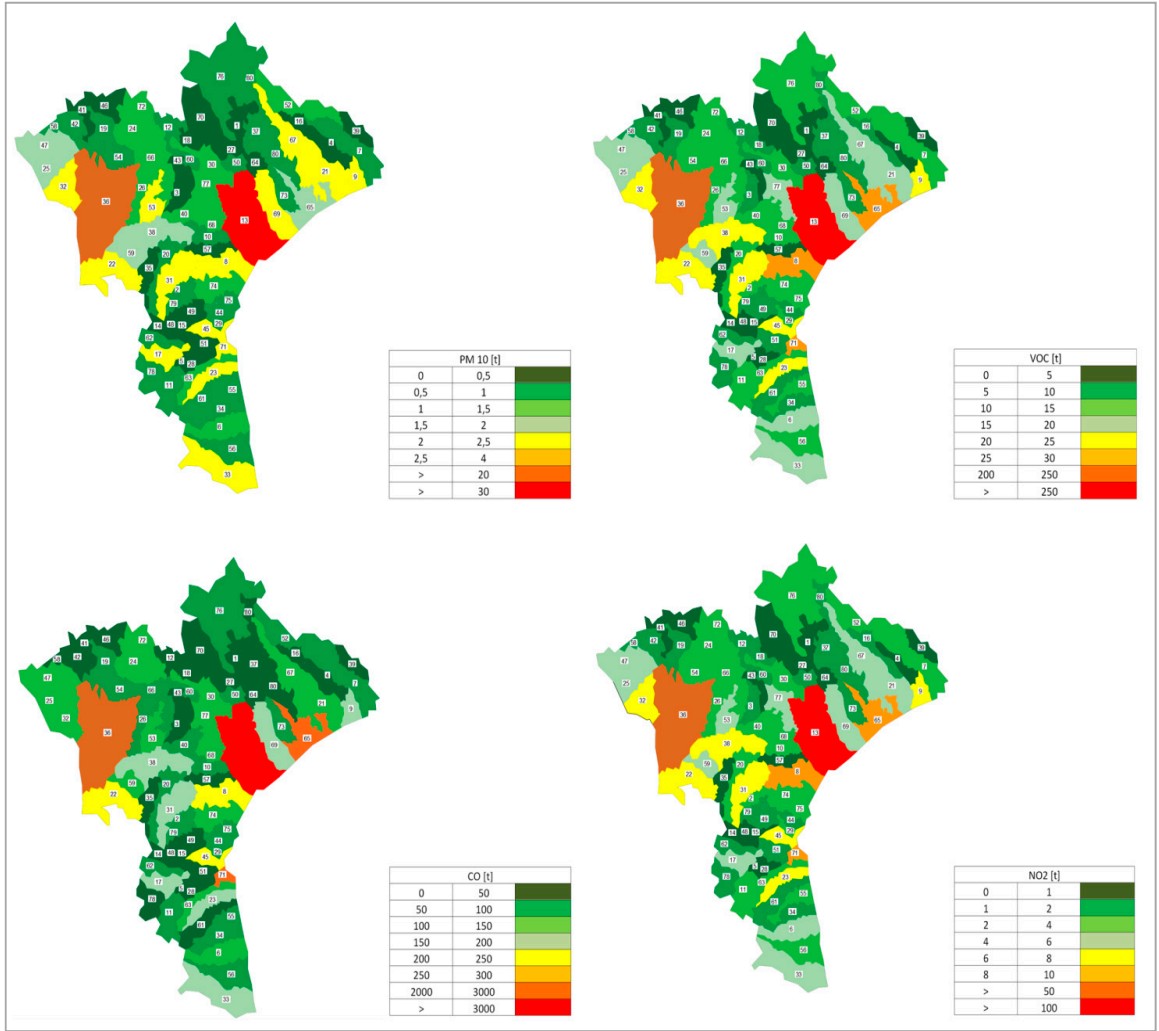

**Figure 5.** Choropleth maps of the eighty municipalities in the province of Catanzaro, relative to the total production in tons, in 2015, of PM (Particulate Matter), VOC (Volatile Organic Compounds), CO (Carbon Oxide), and NO$_2$ (Nitrogen dioxide)

Comparing Figures 5 and 6, it is possible to observe a colorimetric variation in the maps. In particular, the municipalities that were previously those with the largest production of pollutants (Figure 5), had now lost their primacy, and the critical areas in previous maps that did not emerge were now highlighted (Figure 6). This was the case of the municipality of Soverato (*n* = 71 in Table 2), which turned out to be a very polluted municipality that fell into the red zone. This small municipality, after Catanzaro and Lamezia T., had the largest fleet of vehicles (7129), but far from the 74,178 vehicles in Catanzaro. For each municipality in the province of Catanzaro, the emission of the pollutants considered was calculated with respect to the number of vehicles. The data obtained, as expected, were very similar to each other. In particular, the CO[t]/number of vehicles value was around 0.045; the VOC[t]/number of vehicles was around 0.004, the PM[t]/number of vehicles was around 0.0005, and the NO$_2$[t]/ number of vehicles was around 0.0015. All this confirms how the predominant factor is the density of vehicles. From Table 2, it is possible to observe how these data are strongly correlated to vehicle density (number of vehicles/Km$^2$). Soverato is the municipality that has the highest density in the whole province, which was equal to 925 vehicles/km$^2$; Catanzaro instead presented a density of 725 vehicles/km$^2$, and Lamezia Terme had 328 vehicles/km$^2$. These results show how data reported for a smaller area allow for a more precise analysis of the territory. The built choropleth maps turned out to be a valid instrument of investigation and analysis, but above all, for the local authorities responsible

for the control and protection of the territory. These maps show that with current means, it is possible to obtain a lot of indirect information on the territory.

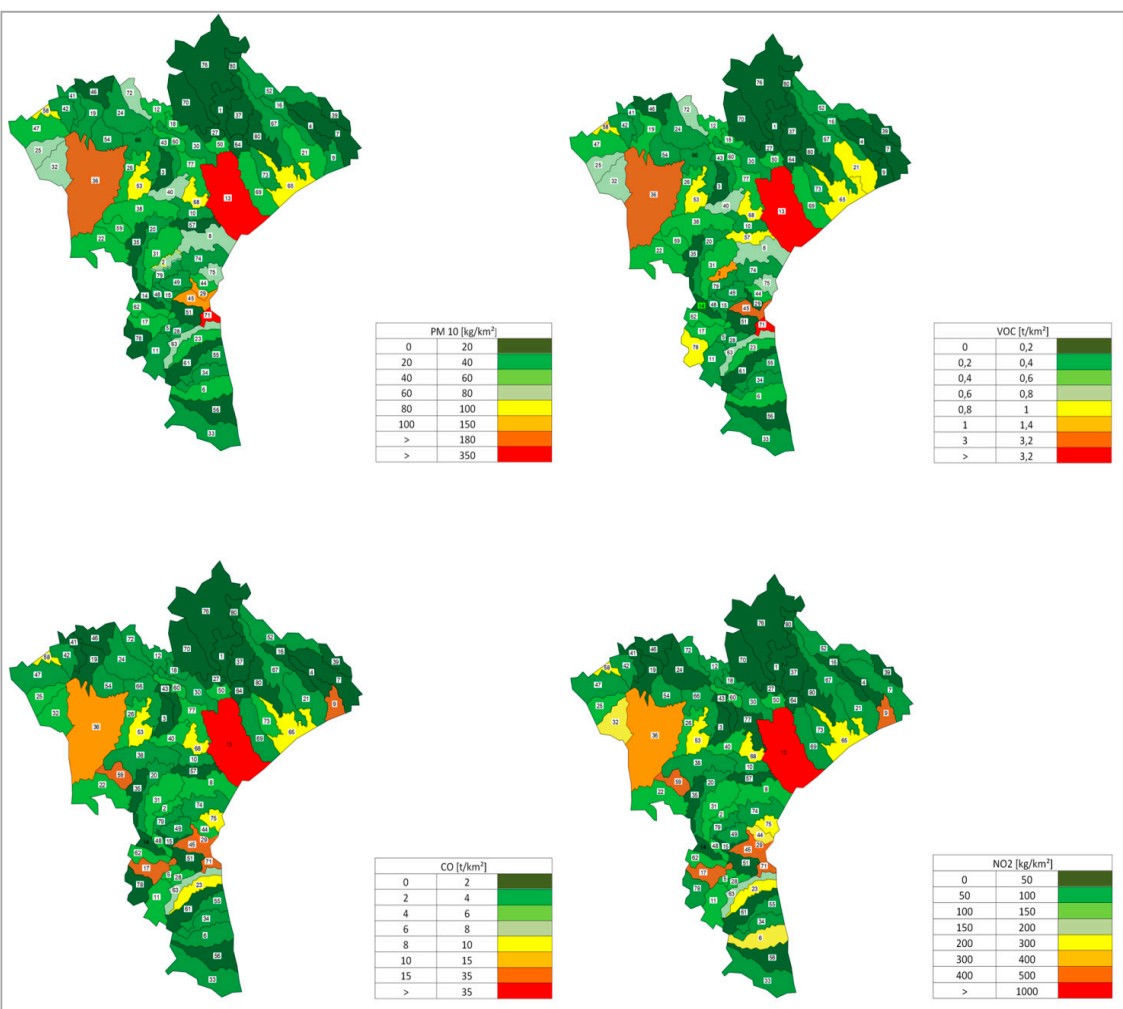

**Figure 6.** Choropleth maps of the eighty municipalities in the province of Catanzaro, relative to the production for area ($Km^2$), in 2015 of PM, VOC, CO, and $NO_2$.

### 3.3. Hypothetical Scenario

The percentage reduction of emissions if all vehicles in the province of Catanzaro were equipped with Euro 0 rather than with Euro 1 engines was evaluated. This choice was dictated by the desire to verify and quantify the reduction in the emission of pollutants that was only assumed by the smallest variation in the type of engines present in the car fleet. So, only a shift toward superior technology was assumed, which should be emphasized as also being obsolete. Table 3 shows the data of real and hypothetical emissions, and the reduction of pollutants in the province of Catanzaro.

**Table 3.** Real and hypothetical emissions and the reduction in pollutants in the province of Catanzaro.

| Pollutants | Real Emissions [t] | Hypothetical Emissions without Euro 0 [t] | Percentage Reduction (%) |
|---|---|---|---|
| CO | 12496.08 | 3637.71 | 70.89 |
| VOC | 1209.01 | 598.69 | 50.48 |
| NMVOC | 1138.99 | 555.69 | 51.21 |
| $CH_4$ | 70.017 | 42.99 | 38.59 |
| NO | 2166.49 | 1495.33 | 30.99 |
| $NO_2$ | 16.10 | 15.24 | 5.86 |
| $N_2O$ | 491.12 | 15.25 | 96.89 |
| $NH_3$ | 59.62 | 68.44 | −14.79 |
| $PM_{2.5}$ | 172.15 | 125.82 | 26.91 |
| $PM_{10}$ | 195.12 | 147.19 | 24.56 |
| $PM_{exhaust}$ | 144.26 | 99.90 | 30.75 |
| $CO_2$ | 626645.30 | 560830.10 | 10.50 |
| $SO_2$ | 3.88 | 3.84 | 1.02 |

The difference between real and supposed emissions showed a significant percentage reduction. Some very harmful pollutants such as carbon monoxide, volatile organic compounds, or particulate matter in this hypothesis had a reduction of 71%, 50%, and about 25%, respectively. Additionally, methane was 38%, nitric oxide was 30%, and the carbon dioxide was 10%. Nitrous oxide emissions could even be reduced by as much as 96%. Other pollutants, even if only by a few percentage points, were less present in the vehicle fumes. It is striking that the emissions of ammonia, rather than decreasing, increased by almost 15 percentage points. This is linked to the fact that cars with Euro 1, unlike the vast majority of Euro 0, are supplied with a catalyst. In reducing nitric oxide emissions during combustion, this produces a certain amount of ammonia. Apart from the increase of the latter (among other things that had small percentages), there was a substantial reduction in emissions. This study highlighted how a minimum improvement in engine technologies, in this case from Euro 0 to Euro 1, would result in a reduction of some important pollutants. In the analyzed car parks, it was confirmed that the Euro 0 vehicles contributed significantly to the quantity of pollutants emitted from vehicular traffic.

## 4. Conclusions

The study showed that it is possible to carry out a preventive screening through the use of new software and to identify areas of potential criticality with regard to vehicular emissions, on which it is subsequently possible to specifically prepare suitable samplings on small-scale territory and to verify the effectiveness of policies that can be adopted for the reduction of pollution. By applying the Corinair methodology, supported by the Copert 4 software, the entire car fleet and its emissions were analyzed respectively.

The emissions from vehicular traffic relating to the CO, VOC, PM, and $NO_2$ pollutants were estimated for each province in Calabria for 2015.

Furthermore, it was possible to create maps related to vehicle emissions on different scales: provincial and municipal, which allows for a more punctual and precise analysis. It has been shown that in the territory of the province of Catanzaro, assuming that the cars in the car park with Euro 0 technology had Euro 1 technology, there would have a significant reduction in pollutants such as 96% of nitrous oxide, 71% of carbon monoxide, and 50% of volatile organic compounds. All this then underlines how it is possible to significantly reduce the emissions by several percentage points by modernizing the vehicle fleet in a small way. Moreover, it is possible to create municipal-scale choropleth maps of the main pollutants present in vehicular emissions. These maps can be a valuable tool for analyzing the territory in relation to emissions from vehicular traffic, and can allow local authorities to direct their political choices aimed toward environmental protection and their citizens

such as the adoption of measures that favor the modernization of car fleets, for example, through the use of incentives.

**Author Contributions:** Conceptualization, P.D.L.; Methodology, S.M.; Software, S.M.; Validation, P.D.L. and A.M.; Formal analysis, S.M.; Investigation, D.A.; Data curation, S.M. and D.A.; Writing—review and editing, P.D.L. and D.A.; Supervision, P.D.L. and A.M. All authors have read and agreed to the published version of the manuscript.

**Funding:** This research received no external funding.

**Conflicts of Interest:** The authors declare no conflicts of interest.

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
