# Peer review of "Vehicular Emission: Estimate of Air Pollutants to Guide Local Political Choices. A Case Study"

_environments, doi:10.3390/environments7050037_

Round 1

Reviewer 1 Report

1. The abstract should be re-written. Not just for style but for content to include the

2. Please provide the reference for the numbers in lines 61 - 62.

3. Please provide a quick description of the beta parameter in line 91.

4. Eliminate figure 1 - screen captures of software are not generally appropriate for journal articles (even a case study) and it's not necessary.

5. Please improve the quality of Figure 2 - perhaps something other than a metalic colour scheme and the font is blurry.

6. Please turn the dark grey background to white in Figure 3.

7. I didn't get why (line 160) the authors decided (with the word "Subsequently" after figure 2) to focus on Catanzaro. Can they elaborate a bit more? It's a good idea but a better segue would help the flow.

8. Table 2 is a lot of data and it's impressive that the data are available at such a descretization. But it would be a better way to convey this data if the authors converted the table to a coloured map showing the grades of vehicle density (number per km2). They need to add a scale bar (also please add UTM or lag/long coordinates to one corner and a north arrow) to Figure 3, and then simply show show the density distribution throughout the region. The maps would likely also infer information on Catanzaro's terrain which is a mix of valleys and small mountains.

9. I would suggest a graded colour scheme for Figure 6 is preferred over a chloropleth which is less intuitive.

Author Response

Dear Reviewer,

thank you for your time spent reviewing our manuscript and for giving us important tips for improving it. We have really appreciated your kind and delicate ways to suggest your comments. The manuscript has been revised in many parts. Many figures have been improved. The changes have been highlighted in yellow.  The answers to your suggestions point by point are shown in the attachment.

We hope that in this form the manuscript will find your approval to be considered for its publication.

We thank you and send you our best regards.

Reviewer 2 Report

Vehicular emission: estimate of air pollutants to guide local political choices. A case study

Sergio Mazza, Donatella Aiello, Anastasia Macario, Pierantonio De Luca*

The paper presents an interesting dataset on the vehicular emission in a southern region of Italy, using a Copert4 tools to estimate the contribution of car fleets in each studied provinces and municipalities. The topic is of high importance and the manuscript aims to help local policy makers to find solutions that have a lower environmental impact.

The authors present their findings quite clearly and the paper is interesting. The used method scientifically sound and is well explained.

I would recommend publishing this paper after addressing some important issues to make the paper stronger and suitable for publication.

Major suggestions:

  • The authors show a significant improvement in the emission information due to vehicular traffic in a region where, according to the authors, there is no data relating to these emissions. Moreover, the proposed data are in line with the purpose of this work. I would ask a curiosity to the authors; why did the authors eliminate only euro zero engines? the other obsolete engines have not been eliminated to obtain information closer to what is the result required by the Treaty of Paris, which has been also mentioned in the introduction.
  • In my opinion, the authors should pay more attention to the insertion of some references such as in line 44 and 97 where the statements shown are not supported by a reference. Furthermore, in some cases, the references for some figures in the text are not shown.

Minor comments and remarks:

Abstract

The abstract makes the reader understand the purpose of the paper but in my opinion, the line 15-19 are slightly confused, I recommend the authors to rewrite it.

Introduction

  • Introduction is generally well written, and it clearly focuses on the problem and stated hypothesis. In my opinion there are a leak of information in the bibliography which need to be integrated. For instance, in a recent study Moretti et al. (Moretti et al. 2020) showed the different impact of natural and anthropic aerosol emission, the latter mainly coming from the vehicular traffic coming from the cities near the sampling sites. While, the potential effect of pollution on biota is missing and should be added to be thorough. Some of these are in fact used as bioindicators of air quality as demonstrated by (Talarico et al. 2014; Giglio et al. 2017). (The full reference details are reported below)
  • In line 54-59 the authors focus attention on a key point in their paper, namely the adoption of European policies to contain pollution emissions, especially on emissions from road traffic. The intention is correct, but their discourse result confusing.

Material and methods

The experimental part is well organized and is easy for the reader to follow. The beta parameter mentioned in line 91 is not well explained both how it is calculated and its use. Furthermore, in the same figure in which the beta parameter is mentioned (Figure 1a) the RVP value is reported, but also this is not explained. In my opinion, it is useful to specify these two parameters to allow the reader to have a complete picture.

Result and discussion

The results illustrated in figure 5 show how the province with lower production of total pollution, on the other hand, has the greatest amount of emissions per inhabitant. In my opinion, this result can be attributable to an "older" fleet in this province relate to the number of inhabitants, consequently the emissions are higher than those of a province that has a newer and more performing fleet.

Conclusions

The authors focused attention on the results obtained in the province of Catanzaro and on the reduction of pollutants by eliminating an obsolete technology such as the euro 0 engines. This part of the conclusions results clear and well organized, but the part concerning the difference between the provinces of the Calabria region has been omitted (reported in section 3.1). I suggest the author to implement the latter one.

Editorial suggestions:

  • Line 54-58: these lines are slightly confused. Please this part needs to be rewritten
  • Line 63-65: we move from European countries to regions and provinces. Specify which regions and provinces, Italian?
  • Line 84: is reported “and also thanks to possibility to enter a series of data.” Please conclude the phrase. Which type of data?
  • Line 90: ….have been entered. Please change with loaded.
  • Line 91: ….Arpa Calabria. Please specify the acronym
  • Line 201: “All the eighty municipalities of the province of Catanzaro were analyzed.” This sentence is redundant and meaningless in this position.
  • Line 228: …each country… Please correct with “each municipality”
  • Line 239: ….. with today's current means,…. Please correct with “with current means”
  • Line 272: ….. the entire car fleet…. Precise which entire car fleet were analyzed

Figure and Tables:

  • Figure 2: Please change “Number of vehicular” with “Number of vehicles”
  • Table 1: Please insert the referred year of the data
  • Table 2: Please change “for city of the province of Catanzaro” with “for cities of Catanzaro province”

Reference:

  • Giglio, A., Ammendola, A., Battistella, S. et al. Apis mellifera ligustica, Spinola 1806 as bioindicator for detecting environmental contamination: a preliminary study of heavy metal pollution in Trieste, Italy. Environ Sci Pollut Res 24, 659–665 (2017). https://doi.org/10.1007/s11356-016-7862-z
  • Moretti, S.; Salmatonidis, A.; Querol, X.; Tassone, A.; Andreoli, V.; Bencardino, M.; Pirrone, N.; Sprovieri, F.; Naccarato, A. Contribution of Volcanic and Fumarolic Emission to the Aerosol in Marine Atmosphere in the Central Mediterranean Sea: Results from Med-Oceanor 2017 Cruise Campaign. Atmosphere 2020, 11, 149. https://doi.org/10.3390/atmos11020149
  • Talarico, P. Brandmayr, P.G. Giulianini, F. Ietto, A. Naccarato, E. Perrotta, A. Tagarelli, A. Giglio Effects of metal pollution on survival and physiological responses in Carabus (Chaetocarabus) lefebvrei (Coleoptera, Carabidae), European Journal of Soil Biology, Volume 61, 2014, 80-89, https://doi.org/10.1016/j.ejsobi.2014.02.003.

Author Response

Dear Reviewer,

thank you for your time spent reviewing our manuscript and for giving us important tips for improving it. The manuscript has been revised in many parts. The changes have been highlighted in yellow.  Answers to your suggestions point by point are shown in the attachment.

We hope that in this form the manuscript will find your approval to be considered for its publication.

We thank you and send you our best regards.
